# NMR Analyses of the Enzymatic Degradation End-Products of Diabolican: The Secreted EPS of *Vibrio diabolicus* CNCM I-1629

**DOI:** 10.3390/md20120731

**Published:** 2022-11-23

**Authors:** Sophie Drouillard, Laurent Poulet, Claire Boisset, Christine Delbarre-Ladrat, William Helbert

**Affiliations:** 1CERMAV, CNRS and Grenoble Alpes Université, 38000 Grenoble, France; 2Ifremer, MASAE, EM3B Laboratory, F-44300 Nantes, France

**Keywords:** exopolysaccharide, diabolican, hyaluronan lyase, NMR

## Abstract

Diabolican, or HE800, is an exopolysaccharide secreted by the non-pathogenic Gram-negative marine bacterium *Vibrio diabolicus* (CNCM I-1629). This polysaccharide was enzymatically degraded by the *Bacteroides cellulosilyticus* WH2 hyaluronan lyase. The end products were purified by size-exclusion chromatography and their structures were analyzed in depth by nuclear magnetic resonance (NMR). The oligosaccharide structures confirmed the possible site of cleavage of the enzyme showing plasticity in the substrate recognitions. The production of glycosaminoglycan-mimetic oligosaccharides of defined molecular weight and structure opens new perspectives in the valorization of the marine polysaccharide diabolican.

## 1. Introduction

Many microorganisms, including marine bacteria, secrete extracellular polysaccharides, called exopolysaccharides (EPSs). The structural diversity of EPSs—as for all other polysaccharides—lies in the stereochemistry of the carbohydrate residues, the glycosidic linkage between the residues, and, in turn, the identity of the repetition units. The structural diversity of marine EPSs as well as the associated biological and physico-chemical properties have been unexplored so far. Although production costs and compliance with regulations hamper the industrial-scale use of marine EPSs facing the already marketed plant and algal polysaccharides [1,2], the production of these macromolecules has several technical advantages. Their production can be easily controlled independently of seasonal variation, for example, and they can be extracted and purified under mild conditions. Therefore, potential valorization of marine EPS is restricted to niche applications in biomedical and cosmetic industry.

In this context, HE800 EPS, or diabolican [3], the EPS secreted by the non-pathogenic Gram-negative marine bacteria *Vibrio diabolicus* (CNCM I-1629), was shown to promote bone reconstruction and improve in vitro skin regeneration [4,5]. These very interesting biological properties were explained by the glycosaminoglycan-like structure of the polysaccharide made of a tetrasaccharide repetition unit [→3)-β-D-GlcNAc-(1→4)-β-D-GlcA-(1→4)-β-D-GlcA-(1→4)-α-D-GalNAc-(1→ ] [6]. In particular, this chemical structure displays similarity with hyaluronic acid (HA), such as an equal ratio of glucuronic acid (GlcA) to *N*-acetyl-hexosamines and presence of the HA disaccharide (Figure 1). A polysaccharide with an identical repeating unit structure is also secreted by another *Vibrio* strain—*Vibrio* sp. MO245 was isolated from bacterial mats found in Moorea Island (French Polynesia) [7] while *Vibrio diabolicus* (CNCM I-1629) was isolated from deep sea [8].

Because the molecular weight of diabolican varies from 0.8 to 1.5 × 10^6^ g.mol^−1^ depending on the production batch, monitoring the size of the oligo- or polysaccharide was envisioned to better control the biological properties, as well as evidencing new functionalities of the macromolecules. A previous work involving screening of commercially available enzymes allowed limited depolymerization of diabolican. In particular, no depolymerizing activity was detected with four commercially available hyaluronidase and hyaluronate lyases; only an endo-α-N-acetyl-galactosaminidase exhibited activity on diabolican [9]. Recently, the screening of a large set of predicted carbohydrate active enzymes (e.g., CAZymes) revealed new enzyme activities and led to the creation of new CAZy families [10]. Among the screened enzymes, we found that the *Bacteroides cellulosilyticus* WH2 hyaluronan lyase (Genbank ALJ61728.1) grouped in the newly created PL33 family, was active in diabolican, providing a new series of oligosaccharides. Herein, we report the full NMR analyses of the four oligosaccharide end-products.

## 2. Results and Discussion

Enzymatic degradation of diabolican leads to the production of a series of oligosaccharides in agreement with the endo-mode of action of the *B. cellulosilyticus* WH2 hyaluronan lyase. The end-products were purified by size-exclusion chromatography (Figure 2) and analyzed in detail by NMR. Knowing the structure of the polysaccharide and that the enzyme cleaves at a GlcA position, giving an unsaturated glucuronic residue at the non-reducing end, the chemical structures of the oligosaccharides were deduced in a straightforward manner. The ^1^H NMR spectrum (Figure 3A) of the smallest oligosaccharide, eluting at a 500 min retention time, presented the characteristic signals of the unsaturated glucuronic residue located at the non-reducing end (Δ4,5UA-H4 = 6.08 ppm; Δ4,5UA-H1 = 5.42 ppm). Integration of the anomeric signals suggested that it was a tri-saccharide. Analyses of 1D (^1^H and ^13^C) and 2D (COSY, HSQC, HMBC) NMR showed the residue located at the reducing end was the N-acetyl-glucosamine (GlcNAc) resonated at GlcNArα-H1 = 5.20 ppm and GlcNArβ-H1 = 4.80 ppm. The proton correlation system of this residue was clearly evidenced on the homonuclear ^1^H–^1^H chemical shift correlated spectrum (COSY), and the chemical shifts are reported in Table 1. As expected, the third residue located between the reducing end and the non-reducing end was *N*-acetyl-galactosamine (GalNAc), for which the anomeric proton resonated at GalNAcr’α-H1 = 5.49 ppm and GalNAcr’β-H1 = 5.51 ppm. Heteronuclear ^1^H–^13^C coupling revealed by the heteronuclear multiple bond correlation (HMBC) spectrum allowed the confirmation of the linkages between the residues giving the trisaccharide chemical structure: Δ4,5UA-β-1,4-GalNAc-α-1,3-GlcNAcrα/β (Figure 3A).

The peak at 510 min elution time in the size exclusion chromatogram was composed of two tetrasaccharides characterized by two distinct unsaturated glucuronic residues at the non-reducing end (Figure 3B). The first, Δ4,5UA1 (Δ4,5UA1-H4 = 6.05 ppm; Δ4,5UA1-H1 = 5.41 ppm), had proton and carbon chemical shifts similar to those reported on the unsaturated residue of the trisaccharide. The HMBC spectrum (Figure 4B) confirmed that it was linked to GalNAc, for which chemical shifts were also conserved, suggesting that this tetrasaccharide had the chemical structure Δ4,5UA-β-1,4-GalNAc-α-1,3-GlcNAc-β-1,3-GlcArα/β. This structure was validated by the complete attribution of the ^1^H and ^13^C chemical shifts by the ^1^H-^13^C correlation, demonstrating the linkages between the residues (Table 1).

Analysis of the proton correlation system revealed that the unsaturated glucuronic residue of the second tetrasaccharide had very different chemical shifts and, notably, a more downfield anomeric proton Δ4,5UA2-H1 = 5.22 ppm. Inspection of the HMBC spectrum (Figure 4B) demonstrated that this residue was linked to the glucuronic acid residue (GlcA-H1 = 4.72 ppm). Considering the degradation modalities of the enzyme and what end-products could be obtained (Figure 5), it was hypothesized that the second tetrasaccharide had the chemical structure Δ4,5UA-β- 1,4-GlcA-β-1,4-GalNAc-α-1,3-GlcNAcrα/β. Inspection of the 1D and 2D NMR spectra confirmed the hypothesis, and the proton and carbon chemical shifts are reported in Table 1. The linkages between residues were demonstrated in the HMBC spectra and are highlighted in Figure 5B.

The longer-size oligosaccharide, eluting at 445 min in the size-exclusion chromatogram, was made with an unsaturated glucuronic acid whose chemical shifts were similar to the trisaccharide and the Δ4,5UA1 of the first tetrasaccharide studied. HMBC analysis confirmed that it was linked to the GalNAc residues, like the other oligosaccharides. The chemical shift of the residue located at the reducing end were similar to the GlcNAc found in the trisaccharide and in the second tetrasaccharide investigated. Inspection of the anomeric region of the ^1^H and ^13^C spectra revealed eight anomeric protons and carbons: GalNAcrα-H1/C1= 5.20/90.89 ppm; GalNAcrβ-H1/C1 = 4.79/94.50 ppm; GalNacr’-H1/C1 = 5.46/97.5 ppm; GlcA1-H1/C1 = 4.75/103.24 ppm; GlcA2-H1/C1 = 4.61/102.28 ppm; GlcNAc-H1/C1 = 4.64/100.42 ppm; GalNAc-H1/C1 = 5.48/97.06 ppm, Δ4,5-UA-H1/C1 = 5.43/100.88 ppm. These anomeric signals were in agreement with the heptasaccharide adopting the α- and β-anomeric configurations. The possible cleavage site of the polysaccharide (Figure 5) explained the production of the heptasaccharide, which could be further degraded in tri- and tetrasaccharide. The complete attribution of the protons and carbons of all the residues as well as the linkages between the residues resembling what was reported for the tri- and tetrasaccharides confirmed the heptasaccharide structure: Δ4,5UA-β-1,4-GalNAc-α-1,3-GlcNAc-β-1,4-GlcA-β-1,4-GlcA-β- 1,4-GalNAc-α-1,3-GlcNAcrα/β.

In-depth analyses of the oligosaccharides produced by the degradation of diabolican by the *B.cellulosilyticus* WH2 hyaluronan lyase confirmed once more the structure of the polysaccharide [6,11]. This polysaccharide, known to mimic glycosaminoglycan, is also subjected to enzymatic degradation using a glycosaminoglycan lyase. The structure of the oligosaccharide end-products demonstrated two sites of cleavage, either the β (1,4) linkage between GlcNAc and GlcA or between two GlcA, showing plasticity of the active site of the enzymes able to accommodate both GlcNAc and GlcA in position -1 of the active site.

Together with enzymes exhibiting a different bond cleavage specificity, such as a *N*-Acetyl-galactosaminidase [9], this *B. cellulosilyticus* hyaluronan lyase constitutes an innovative tool for obtaining glycosaminoglycan-mimetic oligosaccharides of defined molecular weight and structure. These enzymes are important both to achieve fundamental insights into the structure–function relationship and to develop applications in the human health and well-being fields through an optimized enzymatic process.

## 3. Materials and Methods

### 3.1. Production of the Oligosaccharides

*Bacteroides cellulosilyticus* WH2 hyaluronan lyase (genebank ALJ61728.1) was produced and purified according to Helbert et al. [10]. The gene encoding the enzymes was cloned in pHTP1 vector (kanamycin resistant) with an N-terminal poly-histidine tag for purification. The plasmid was transformed in BL21(DE3) pLys S strain and grown in NZY auto-induction LB media (Nzytech, Portugal) at 25 °C. The proteins were purified by affinity chromatography using a 1 mL HisTrap™ HP column (GE Healthcare) connected to an NGC chromatography system (Bio-Rad).

Oligosaccharides were produced by incubating 10 mL HE800 EPS (0.1% *w*/*v* in 100 mM Tris- HCl pH 7.5) with 200 µL of the purified enzyme (2.45 µg/µL) at 25 °C overnight. Then, the oligosaccharides were purified by semi-preparative gel permeation chromatography using three HiLoad^®^ 26/600 Superdex^®^ 30 pg (GE Healthcare) columns mounted in series and connected to a semi-preparative size-exclusion chromatography system, which consisted of a Knauer pump (pump model 100), a refractive detector (iota2 Precision instrument) and a fraction collector (Foxy R1) mounted in series. The elution was conducted at a flow rate of 1.2 mL.min^−1^ at room temperature using 100 mM (NH_4_)2CO_3_ as eluent. The collected fractions were freeze-dried prior to NMR analyses.

### 3.2. NMR

Carbon-13 and proton NMR spectra were recorded with a Bruker Avance 400 spectrometer operating at a frequency of 100.618 MHz for ^13^C and 400.13 MHz for ^1^H. Samples were solubilized in D_2_O (pH = 6.5) and analyzed at a temperature of 293 K. Residual signal of the solvent was used as internal standard: HOD at 4.85 ppm at 293 K. ^13^C spectra were recorded using 90° pulses, 20,000 Hz spectral width, 65,536 data points, 1.638 s acquisition time, 1 s relaxation delay and between 8192 and 16,834 scans. Proton spectra were recorded with a 4006 Hz spectral width, 32,768 data points, 4.089 s acquisition times, 0.1 s relaxation delays and 16 scans. The 1H and ^13^C-NMR assignments were based on ^1^H-^1^H homonuclear and ^1^H-^13^C heteronuclear correlation experiments (correlation spectroscopy, COSY; heteronuclear multiple-bond correlation, HMBC; heteronuclear single quantum correlation, HSQC). They were performed with a 4006 Hz spectral width, 2048 data points, 0.255 s acquisition time, and 1 s relaxation delay; 32 to 512 scans were accumulated.

## Figures and Tables

**Figure 1 marinedrugs-20-00731-f001:**
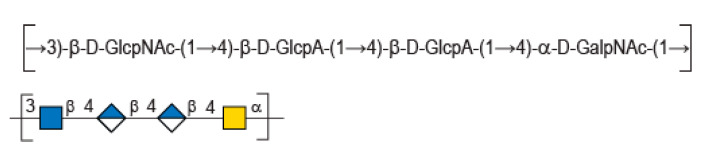
Chemical structure of diabolican repeating unit.

**Figure 2 marinedrugs-20-00731-f002:**
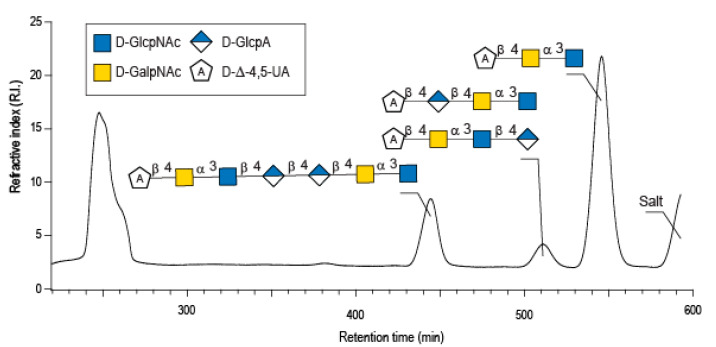
Purification by size exclusion chromatography of the enzymatic degradation products of diabolican incubated with the *B. cellulosilyticus* WH2 hyaluronan lyase. Inset: the structure of the oligosaccharides determined by NMR analyses.

**Figure 3 marinedrugs-20-00731-f003:**
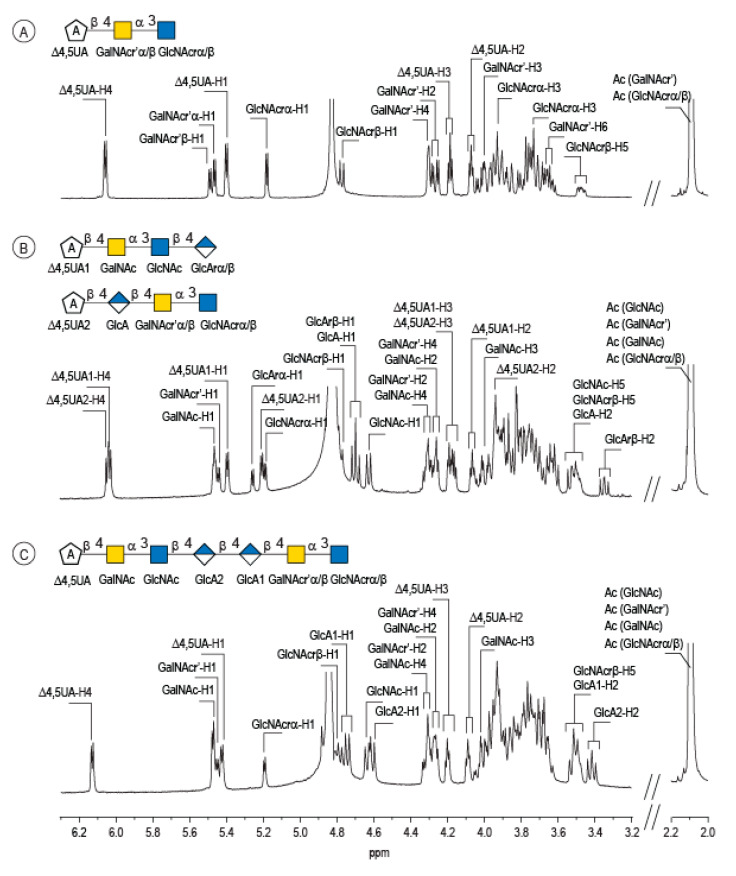
Characterization of the enzymatic degradation products of diabolican by the *B. cellulosilyticus* WH2 hyaluronan lyase. ^1^H NMR spectra of the trisaccharide (**A**), tetrasaccharides (**B**) and heptasaccharide (**C**) isolated after purification by chromatography. The spectra were recorded at 293 K.

**Figure 4 marinedrugs-20-00731-f004:**
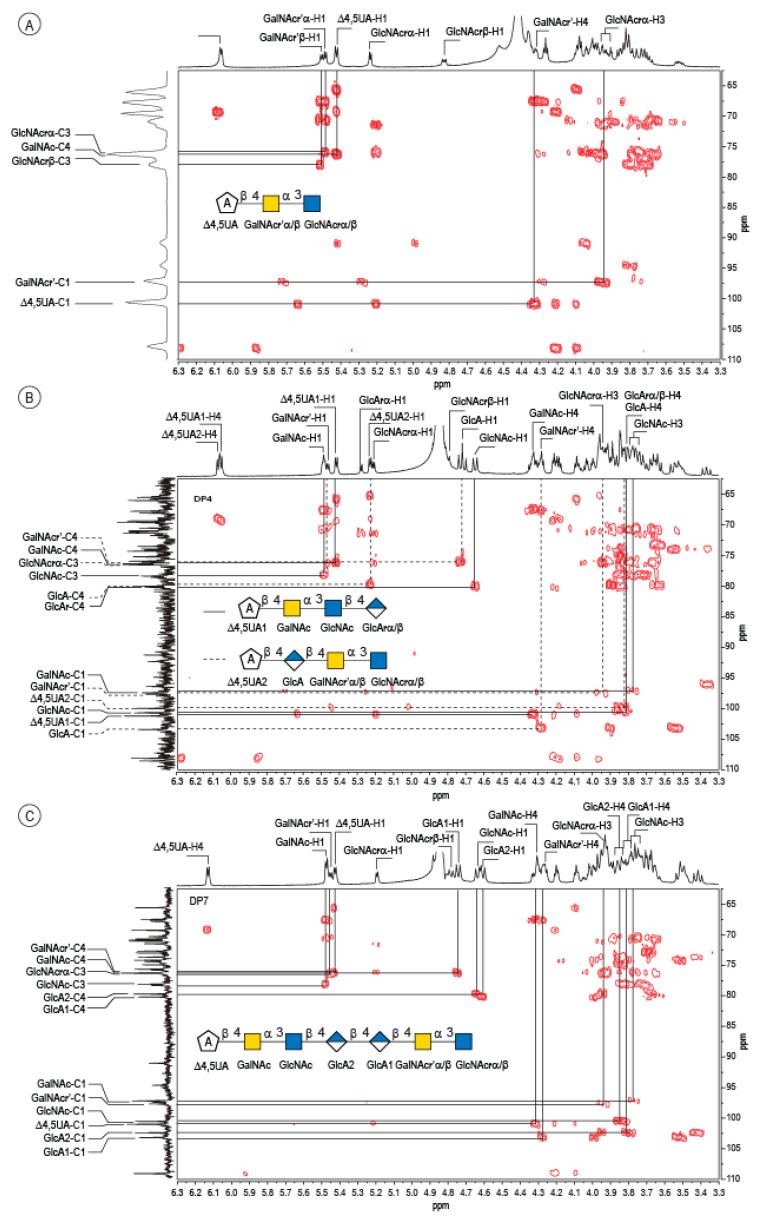
4: Analyses of the oligosaccharides by NMR. Details of the HMBC spectra recorded on the DP3 (**A**), the mixture of two DP4 (**B**) and the DP7 (**C**). The spectra were recorded at 293 K.

**Figure 5 marinedrugs-20-00731-f005:**
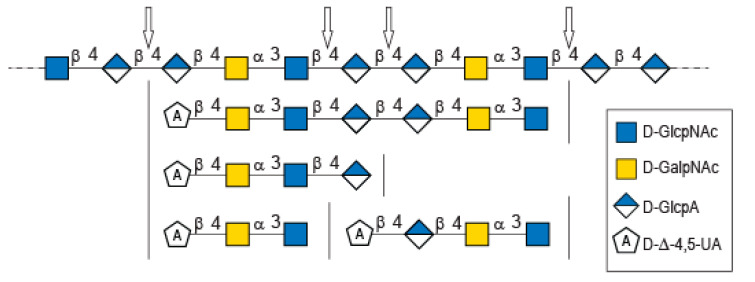
Possible cleavage sites of diabolican by the *B. cellulosilyticus* WH2 hyaluronan lyase identified from the oligosaccharide obtained after enzymatic degradation.

**Table 1 marinedrugs-20-00731-t001:** ^1^H and^13^C NMR chemical shifts recorded on diabolican oligosacharides obtained after digestion of the polysaccharide by *B. cellulosilyticus* WH2 hyaluronan lyase.

Sugar Residue		1	2	3	4	5	6 (6a,6b)	Ac (CH3,CO)
**DP3**								
Δ**4,5-UA:** D-UA-(1→	^1^H	5.42	4.09	4.21	6.08	no	no	
	^13^C	100.89	69.25	65.55	108.20	143.20	168.25	
**GalNAcr’**α/β: →4)-α-D-GalNAc*p*-(1→	^1^H	5.49/5.51 ^a^	4.29	4.01	4.33	3.96	3.67, 3.77	2.10
	^13^C	97.35/97.11 ^a^	49.62	67.48	76.15	70.60	59.95	21.88, 174.43
**GlcNAc** α: →3)-α-D-GlcNAc*p*	^1^H	5.20	4.04	3.95	3.73	3.91	3.77, 3.93	2.12
	^13^C	90.89	52.36	75.78	70.92	71.23	60.29	22.09, 174.21
**GlcNAc** β: →3)-β-D-GlcNAc*p*	^1^H	4.80	3.78	3.76	3.73	3.50	3.77, 3.93	2.11
	^13^C	97.47	55.07	77.92	70.92	75.65	60.29	22.09, 174.21
**DP4**								
Δ**4,5-UA:** D-UA-(1→	^1^H	5.22	3.95	4.18	6.07	no	no	
	^13^C	99.62	69.00	65.15	108.20	142.98	167.94	
**GlcA:** →4)-β-D-GlcA*p*-(1→	^1^H	4.72	3.54	3.65	3.85	3.88	no	
	^13^C	103.07	73.13	73.92	79.66	74.82	173.00	
**GalNAc:** →4)-α-D-GalNAc*p*-(1→	^1^H	5.46/5.48 ^a^	4.33	3.95	4.28	3.92	3.67, 3.88	2.10
	^13^C	97.28/97.04 ^a^	49.93/50.04	67.73	75.96	70.18	59.75 ^d^	21.85, c value
**GlcNAcr**α: →3)-α-D-GlcNAc*p*	^1^H	5.20	4.05	3.95	3.73	3.91	3.77, 3.96	2.12
	^13^C	90.89	52.39	76.18	70.87	71.26	60.31	22.06, c value
**GlcNAcr**β: →3)-β-D-GlcNAc*p*	^1^H	4.79	3.79	3.76	3.73	3.52	3.77, 3.96	2.12
	^13^C	94.52	55.18	78.22	70.87	75.65	60.16	22.19, c value
**DP4**								
Δ**4,5-UA:** D-UA-(1→	^1^H	5.41	4.08	4.21	6.05	no	no	
	^13^C	100.86	69.26	65.67	107.77	143.48	168.35	
**GalNAc:** →4)-α-D-GalNAc*p*-(1→	^1^H	5.48	4.28	4.01	4.33	3.96	3.67, 3.88	2.10
	^13^C	97.05	49.57	67.41	76.09	70.58	59.82 ^d^	21.85, c value
**GlcNAc:** →3)-β-D-GlcNAc*p*-(1→	^1^H	4.64	3.84	3.76	3.73	3.52	3.77, 3.96	2.12
	^13^C	100.29/100.35 ^b^	53.88	77.98	70.78	75.55	60.16	22.43, c value
**GlcA** α: →4)-α-D-GlcA*p*	^1^H	5.27	3.66	3.83	3.83	4.20	no	
	^13^C	91.97	73.92	70.95	79.84	71.35	173.00	
**GlcA** β: →4)-β-D-GlcA*p*	^1^H	4.70	3.37	3.64	3.83	3.84	no	
	^13^C	96.02	73.56	73.73	79.84	75.89	173.00	
**DP7**								
Δ**4,5-UA:** D-UA-(1→	^1^H	5.43	4.08	4.21	6.12	no	no	
	^13^C	100.88	69.17	65.48	108.96	142.54	167.59	
**GalNAc:** →4)-α-D-GalNAc*p*-(1→	^1^H	5.48	4.28	4.01	4.32	3.95	3.67, 3.88	2.09
	^13^C	97.06	49.58	67.39	76.20	70.59	59.98 ^e^	21.85, 174.56 ^f^
**GlcNAc:** →3)-β-D-GlcNAc*p*-(1→	^1^H	4.64	3.83	3.77	3.73	3.50	3.77, 3.95	2.10
	^13^C	100.42	53.93	77.98	70.77	75.55	60.14	22.41, 174.44
**GlcA2:** →4)-β-D-GlcA*p*	^1^H	4.61	3.42	3.70	3.85	3.94	no	
	^13^C	102.28	72.58	73.56	79.61	74.83	172.70	
**GlcA1:** →4)-β-D-GlcA*p*	^1^H	4.75	3.52	3.72	3.80	3.99	no	
	^13^C	103.24	72.90	74.08	80.09	74.29	173.79	
**GalNAcr’:** →4)-α-D-GalNAc*p*-(1→	^1^H	5.46/5.48 ^a^	4.33	3.95	4.28	3.91	3.67, 3.88	2.09
	^13^C	97.56/97.28 ^a^	50.04/49.93	67.64/673 ^a^	76.16	70.21	59.86 ^e^	21.85, 174.53 ^f^
**GlcNAcr**α: →3)-α-D-GlcNAc*p*	^1^H	5.20	4.04	3.95	3.73	3.91	3.77, 3.93	2.10
	^13^C	90.89	52.35	75.69	70.94	71.36	60.31	21.96, 174.21
**GlcNAcr**β: →3)-β-D-GlcNAc*p*	^1^H	4.79	3.78	3.76	3.73	3.50	3.77, 3.93	2.10
	^13^C	94.50	55.19	78.21	70.94	75.55	60.14	22.19, 174.44

^a^: C→GlcNAcrα/C→GlcNAcrβ; ^b^: C→GlcA α/C→GlcA β; non affected c values: 174.43, 174.51, 174.60, 174.74 ppm; ^d^, ^e^, ^f^: may be interchanged.

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
