# Peer review of "NMR Analyses of the Enzymatic Degradation End-Products of Diabolican: The Secreted EPS of Vibrio diabolicus CNCM I-1629"

_marinedrugs, 2022, doi:10.3390/md20120731_

Round 1

Reviewer 1 Report

Title: What does NMR exactly mean, and I cannot find through the paper.

Abstract: What key problems did this study urgently solve? Please describe more clearly, directly, and logically to impress the readers.

Line 8-9. The phrase was too long and confused for me to understand, and please re-organize it.

Line 9. The name Vibrio diabolicus should be Italic, and please check through the paper.

Line 13. Is , showing better than which shows?

Line 40. How about , in contrast?

Lines 42-44. The phrases were confused too, please re-organize them.

Line 48. suggestion: the HE800 EPS. the

Reviewer 2 Report

The manuscript covers an NMR-spectroscopic study of oligosaccharides obtained from the marine bacterial exopolysaccharide diabolican upon treatment with a bacterial hyaluronan lyase. Since mixtures of structurally defined oligosaccharides could be obtained by size exclusion chromatography, the approach opens potential future utilization options. The structural work has been convincingly carried out and only minor amendments are suggested

The symbols of the hexenuronic and glucuronic acid have been mixed up in the inserts of Figure 1 and Figure 5

Line 68: NAr should be labeled as the beta-anomer at the reducing end

Since pH has a distinct effect on chemical shifts, the pD of the NMR solution should be given

Line 147: Authors indicate mass spectrometry analyses but no data were given. Please clarify

Figure 1 shows an apparent peak arising at ~600 min. Does this correspond to a disaccharide fraction? The authors should comment on this and show the full elution profile

Figure 4, spectrum A: The 13C-spectrum rather than the internal projection should be shown. The signal of H-4 of the hexenuronic acid should be labeled in the 1H projection.

Line 108: the sentence starting with….allowed evidence seven anomeric… should be rephrased

e.g. …provided evidence of seven anomeric….

Typos:

List of Authors: Affiliation of Helbert should be given as a superscript (also  for 1H in line 156)

Line 87: cellulosilyticus

Line  146: Subscripts to be used for ammoniumcarbonate

Line 147:…prior to NMR….
